# Molecular and Cellular Regulations in the Development of the Choroidal Circulation System

**DOI:** 10.3390/ijms24065371

**Published:** 2023-03-11

**Authors:** Satoshi Imanishi, Yohei Tomita, Kazuno Negishi, Kazuo Tsubota, Toshihide Kurihara

**Affiliations:** 1Laboratory of Photobiology, Keio University School of Medicine, 35 Shinanomachi, Shinjuku-ku, Tokyo 160-8582, Japan; 2Department of Ophthalmology, Keio University School of Medicine, 35 Shinanomachi, Shinjuku-ku, Tokyo 160-8582, Japan; 3Tsubota Laboratory Inc., 34 Shinanomachi, 304 Toshin Shinanomachi Ekimae Building, Shinjuku-ku, Tokyo 160-0016, Japan

**Keywords:** choroidal development, vascular development, ocular disorders

## Abstract

Disorders in the development and regulation of blood vessels are involved in various ocular disorders, such as persistent hyperplastic primary vitreous, familial exudative vitreoretinopathy, and choroidal dystrophy. Thus, the appropriate regulation of vascular development is essential for healthy ocular functions. However, regulation of the developing choroidal circulation system has not been well studied compared with vascular regulation in the vitreous and the retina. The choroid is a vascular-rich and uniquely structured tissue supplying oxygen and nutrients to the retina, and hypoplasia and the degeneration of the choroid are involved in many ocular disorders. Therefore, understanding the developing choroidal circulation system expands our knowledge of ocular development and supports our understanding of ocular disorders. In this review, we examine studies on regulating the developing choroidal circulation system at the cellular and molecular levels and discuss the relevance to human diseases.

## 1. Introduction

Disorders in the development and regulation of blood vessels are involved in various ocular disorders. For example, failure in the regression of the hyaloidal artery causes persistent hyperplastic primary vitreous associated with severe and complex symptoms, such as intraocular hemorrhage, cataracts, and retinal detachment [1]. Familial exudative vitreoretinopathy is led by incomplete vascularization in the retina, resulting in the formation of an avascular region in the peripheral retina [2]. Congenital choroidal dystrophies show progressive chorioretinal atrophy. Hence, they lead to progressive vision loss [3,4,5]. The dysregulation of blood vessels, such as choroidal neovascularization, is a destructive symptom not only in congenital disorders but also in age-related macular degeneration [1]. Thus, the appropriate regulation of vascular development is essential for healthy ocular functions. In this review, the regulation of vascular development in the eye, mainly in the choroid, is overviewed.

Before overviewing the choroidal development, we highlight the general development of the ocular system, the embryonic blood supply to the eyes, and the regulation of intraocular vascularization to help understand the development of the choroid.

## 2. The General Development of the Ocular System

The first sign of eye development is the formation of the optic sulcus on the inner surface of the neural ectoderm on each side of the forebrain at 3 weeks of gestation (WG) in humans [6] and at embryonic day (E) 8.25 in mice [7] (Table 1). At up to 3.5 WG in humans and E 8.5 in mice, the optic sulcus projects toward the surface ectoderm and is enhanced to form optic vesicles. The surface ectoderm facing the distal optic vesicle thickens to become the lens placode. Then, the ventral part of the optic vesicle begins to invaginate to form an optic cup, and lens placode invagination forms a lens pit by 5 GW and E 10 in humans and mice, respectively. At the same time, the hyaloidal artery is enveloped by an optic stalk, which connects the optic cup to the brain neural ectoderm. The inner layer of the optic cup differentiates into the neural retina, and the outer layer becomes the retinal pigmented epithelium (RPE). The lens vesicle develops from the lens pit by pinching off the surface ectoderm. The gap between the surface ectoderm and lens vesicle is filled with mesenchymal cells [8,9]. At 8 GW in humans and E 14.5 in mice, the mesenchymal cells touching the lens vesicle form the corneal endothelium. Early studies using chicken embryos suggested that the mesenchymal cells contributing to corneal endothelium derive from neural crest cells [10,11], while a more recent study using lineage-marked transgenic mice showed that the corneal endothelium includes derivatives from neural crest cells and mesodermal cells [12]. Thus, the origin of the corneal endothelium is controversial and should be clarified in future research. The surface ectoderm covered lens becomes a multi-layered epithelium, i.e., the corneal epithelium, at 16 GW in humans [13]. In mice, the formation of the multi-layered corneal epithelium occurs at E 15.5 [13]. The epithelium of the iris and ciliary body develops from the peripheral margin of the optic cup at around 18 GW in humans [14] and E 17 [15] in mice. Iris stroma and ciliary body muscles are differentiated from mesenchymal cells, which are derived mostly from neural crest cells and minorly from the mesoderm [16], while pupil muscles are differentiated from neural ectodermal cells of the optic cup [15].

Not only neural and surface ectoderms but also neural crest cells contribute to ocular development [8]. Neural crest cells are multipotent cells derived from crests of neural folds and contribute to the development of many organs, such as the eyes, skin, dorsal root ganglia, heart, carotid body, bone marrow, teeth, palate, and oral mucosa [17]. The neural crest cells contributing to ocular and periocular development appear in the rostral diencephalon and the metencephalon at around 3 GW in humans and E 8.5 in mice [8]. They migrate to the optic vesicles at 4 GW and E 9.5 in humans and mice, respectively, and fill the space between the surface ectoderm and optic vesicles, except for the anterior segment of optic vesicles where the neural ectoderm touches the surface ectoderm. As the neural crest cells restrict the intrinsic potential of the surface ectoderm to differentiate into the lens, only the surface ectoderm at the anterior segment of the optic vesicles is allowed to form a lens. A possibility that neural crest cells regulate the differentiation of RPE cells has been shown in chickens. When optic vesicle explants are cultured without periocular mesenchyme, the expression of RPE-specific genes is not induced. Thus, appropriate migration of neural crest cells is indispensable in early ocular development. Before lens vesicle formation, the neural crest cells lose the specific marker expression and change to a fibroblast-like morphology. After lens vesicle formation, they fill the gap between the lens and surface ectoderm and contribute to anterior ocular segment formation together with mesoderm-derived mesenchymal cells, as we mentioned above. The pericytes covering retinal and choroidal vessels are also derived from neural crest cells. Although the choroid includes other neural crest derivatives, such as stromal cells and melanocytes, neural crest cells do not differentiate into melanocytes in the ocular region. Melanocytes are differentiated in other sites and migrate into the choroid in late gestation at 27 GW in humans and E 15.5 in mice [18].

The periocular organs, such as the eyelids, conjunctiva, and lacrimal glands, are formed by the interaction between the skin epithelium derived from the surface ectoderm and the periocular mesenchyme, including neural crest derivatives and mesoderm derivatives. The primitive eyelids appear as protruding ridges of the skin epithelium at the cornea periphery at 6 GW in humans [19] and E 11.5 in mice [20]. The protruding epithelium proliferates rapidly and completely covers the cornea at 8 GW in humans and E 16.5 in mice [21]. The mesenchymal cells form muscles and glands in the eyelids. The backside of the eyelids becomes the conjunctiva epithelium. At 6.5 GW in humans and E 13.5 in mice, the conjunctiva epithelium thickens due to the signal from mesenchymal cells condensed in the prospective gland area [22,23]. The thickened epithelium invaginates, elongates, and branches in the stroma to form a lacrimal grand and maturates around 15 GW in humans and around birth in the mouse. In humans, the eyelids start to separate at 20 GW and fully form at 36 GW [19]. The separation of eyelids occurs 10~12 days after birth in mice [21].

## 3. The Embryonic Blood Supply to the Eyes

Because the choroid is an important part of the ocular circulation system, understanding the blood supply to the developing eye would help to understand choroidal development. In humans, the primitive maxillary artery, a facial branch of the primitive internal carotid artery, supplies blood to the optic vesicle up to 30 gestation days (GD), when the primitive ophthalmic artery is formed [24]. After 31 GD, blood supply depends on two branches of the primitive ophthalmic artery, i.e., the primitive dorsal and primitive ventral ophthalmic artery (pDOA and pVOA, respectively) (Figure 1). At about 36 GD, the distal part of the pVOA forms the common nasal ciliary artery, and the pDOA creates the common temporal ciliary artery and the hyaloidal artery. At around 45 GD, the proximal segment of the common nasal ciliary artery fuses with the common temporal ciliary artery and the hyaloidal artery. The rest of the common nasal ciliary artery forms the medial posterior ciliary artery, and the common temporal ciliary artery becomes the lateral posterior ciliary artery [25]. The branches raised from these ciliary arteries, called short and long posterior ciliary arteries, pass through the sclera at the position around the optic nerve entry. The short and long posterior ciliary arteries supply blood to the choroid and iris and ciliary body, respectively. Until 50 GD, the hyaloidal artery takes charge of the blood supply to the intraocular system.

Below, we summarize the changes in the embryonic blood supply to the eyes (Figure 1). However, the facial arteries are formed through complex processes involving branching, anastomosing, and fusion. For example, the formation process of the persistent ophthalmic artery is still controversial. A further understanding of the development of facial arteries would provide further insight into the development of the optic system.

## 4. Overview of Intraocular Vascularization

Vascularization progresses by angiogenesis in the vitreous and the retina. Intraocular vascularization begins from the entry of the hyaloidal artery into the optic cup [1,24]. The hyaloidal artery rapidly extends through the primary vitreous and reaches the posterior pole of the developing lens to supply oxygen and nutrients to intraocular tissues. In mice, the hyaloidal vasculature immediately regresses after birth. This regression is regulated by secreting factors from macrophages and the titration of vascular endothelial growth factor A (VEGFA), the central promotor of vascular formation by neurons [26,27]. The development of the retinal circulation system starts following the regression of hyaloidal vasculature to take over its functions [28,29,30] (Figure 2a). During the first week after birth, the retinal vascular reaches the innermost surface of the retina, the ganglion cell layer, forming the superficial vascular plexus according to the guidance by astrocytes producing VEGFA and extracellular matrix but is not allowed to enter the neuroretina because of titrating VEGFA by the retinal neurons [31] (Figure 2b). The regression of the hyaloidal artery and the development of retinal vessels are also regulated by light response via the stimulation of atypical opsins [32,33]. Afterward, the deep and intermedia vascular plexus is formed up to 3 weeks after birth [34,35]. Thus, regulating VEGFA concentration by retinal neurons plays a role in the development and regression of the hyaloid and retinal vasculatures.

## 5. Anatomy of the Choroid

The choroidal circulation system is also responsible for healthy ocular function. The choroid is a vascular-rich tissue consisting of the outer part of the Bruch’s membrane, choriocapillaris, the Sattler’s layer, the Haller’s layer, and the suprachoroid from the inner to the outer side [36]. The inner part of the Bruch’s membrane attaches to the retinal pigment epithelium (RPE), the outermost layer of the retina, as the basement membrane of the RPE, and the choriocapillaris supplies oxygen and nutrients to the outer layers of the retina, such as the RPE and photoreceptors through the Bruch’s membrane (Figure 2b). Sattler’s and Haller’s layer involves middle and large diameter vasculatures connecting to the choriocapillaris, which are the branches of posterior ciliary arteries or vortex veins. The choriocapillaris forms a very dense capillary bed with lobular patterning (Figure 2c) and consists of fenestrated capillaries on the side facing the RPE to effectively exchange gases, nutrients, and wastes. The appropriate regulation of vascular development is indispensable for the choroidal development to form a unique capillary network. The first sign of choriocapillaris formation in human development appears as the islets of nucleated acidophilic cells in the periocular stroma at 6 WG. A primitive vascular plexus is identified at 12 WG in humans [37,38] and at E 11.5 in mice [1,39] and is organized into a complex network thereafter.

## 6. Cellular and Molecular Regulation of Choroid Development

Studies using genetically modified mice have steadily uncovered cellular and molecular regulations in choroidal development. Since some of them are relevant to human ocular disorders, summarizing these studies could support understanding human ocular disorders further. Hereafter, we overview the studies using genetically modified mice divided by cellular regulation.

## 7. RPE Cells

Some studies that inhibit the differentiation of RPE cells have demonstrated the essential role of the RPE in choroidal development (Table 2). The *Tyrp1-Fgf9* mice ectopically express *Fgf9* in RPE cells under the control of the *Tyrp1* promotor, resulting in the conversion of most RPE cells into the neural retina [40]. In *Tyrp1-Fgf9* mice, choroidal development is inhibited, except for the small region in which differentiation of RPE cells is allowed. The *βB1-Tgfb1* transgenic mice, who produce active transforming growth factor beta 1 (TGFβ1) under the control of the chicken *βB1-crystallin* promotor, show a lack of RPE differentiation which causes choriocapillaris atrophy after birth [41]. Interestingly, the development of choriocapillaris from E 13.5 to E 17.5 does not differ between *βB1-Tgfb1* transgenic and wild-type mice. Hence, RPE is the key regulator of choroidal development. The molecular mechanisms underlying the regulation of choroidal development also support the importance of RPE (Table 2). The role of VEGFA produced by RPE cells is well-researched in choroidal development. The doxycycline-inducible and RPE-specific deletion of *Vegfa* in *Best1-rtTA;TRE-Cre;Vegfa^flox/flox^* mice at the middle gestational stage from E10 to E15, but not at later than E16, induces a lower density in both the choriocapillaris and larger choroidal vessels after birth [42]. *Tyrp1-Cre^tg/0^;Vegfa^flox/flox^* mice and *Best1-Cre;Vegfa^flox/flox^* mice (*Vegfa^RPE^* KO mice), who lack the *Vegfa* gene specifically in RPE cells, show hypoplasia of the choriocapillaris [43,44]. *Vegfa^RPE^* KO mice also exhibit the phenotypes of myopia, such as the elongation of axial length and decreased refraction at 4 weeks after birth [44]. Contrary to *Vegfa^RPE^* KO mice, vasodilatation is observed in *Best1-Cre;Vhl^flox/flox^* mice, showing increased expression of *Vegfa* in RPE cells. Additionally, we reported that the RPE-specific deletion of *Vegfa* in adult mice emaciates the choriocapillaris and results in vision loss [45,46]. The fenestration of choriocapillaris is also strongly regulated by VEGFA secreted by RPE [47,48]. These roles of VEGFA might be mediated by the VEGF receptor 2 (VEGFR2) expressed in the choriocapillaris on the side facing the RPE in mice [49] and humans [50]. Since VEGFR3 shows a similar expression pattern to VEGFR2 in humans [50], VEGFR3 might be involved in the maintenance of the choriocapillaris in humans. The choriocapillaris of the mice expressing a dominant negative *Fgfr1* in RPE cells fails to grow completely and remains immature, indicating that the maturation of the choriocapillaris needs basic fibroblast growth factor (bFGF; also known as FGF2) from the RPE [39]. Thus, the RPE regulates the healthy development and maintenance of the choroid through the secretion of proangiogenic factors such as FGF2 and VEGFA. The dysregulation of proangiogenic factors could be involved in severe myopia.

The dysregulation of *Vegfa* expression in RPE cells may be relevant to human congenic disease (Table 2). The LRP2 mutation is responsible for the Donnai–Barrow syndrome (DB), characterized by agenesis of the corpus callosum, congenital diaphragmatic hernia, facial dysmorphology, ocular anomalies, sensorineural hearing loss, and developmental delay. Most patients with DB exhibit severer myopia and proptosis [51,52]. *Lrp2*-mutant (Lrp2^267/−^) mice also exhibit ocular anomalies, including a thin retina, hyperplasia of the non-pigmented epithelium of the ciliary body, and enlarged and exophthalmic eyes with great elongation of axial length, which is a sign of severe myopia [53]. Interestingly, we found that the *Best1-Cre;Lrp2^flox/flox^* (*Lrp2^RPE^* KO) mice, who lack the *Lrp2* gene specifically in the RPE, reproduce most of the morphological changes in the eyes from the *Lrp2^267/−^* mice, i.e., thin retina, enlarged eyes, and elongation of axial length [44]. In addition to such changes, the thinner choroid and the thinner sclera are reported in *Lrp2^RPE^* KO mice. The thinning of the choroid in *Lrp2^RPE^* KO mice is accompanied by the decreased expression of Vegfa in RPE cells and is due to severe hypoplasia of the choriocapillaris as well as *Vegfa^RPE^* KO mice. In neural retina-specific *Lrp2* deletion mice (*Chx10-Cre;Lrp2^flox/flox^* mice), enlarged eyes, axial length elongation, and hypoplasia of the choriocapillaris are not observed [44]. Therefore, the LRP2-VEGFA axis in RPE cells largely contributes to the development of the choriocapillaris. A part of ocular phenotypes, such as myopia and retinal dystrophy in DB, might be caused by an anomaly in choroidal development.

Matrix metabolizing factors are also essential regulators in vascular formation [54]. Their role in choroidal development has been revealed (Table 2). Tissue inhibitor of metalloproteinase 3 (TIMP3) is produced by RPE cells and inhibits extracellular matrix remodeling, inhibiting vascular remodeling [55]. In *Timp3* KO mice, the choroid shows strong gelatinolytic activity, and the choroidal vessels show dilated morphology, while the morphology of retinal vessels is normal [55]. Surprisingly, the lack of *Timp3* induces the autophosphorylation of VEGFR2 and the phosphorylation of extracellular signal-regulated kinase 1/2 (ERK1/2) in aortic endothelial cells. Because this model lacks *Timp3* in the whole body, it is difficult to discuss the contribution of TIMP3 produced in RPE cells to choroidal development. However, *Timp3* might regulate choroidal development by the regulation of matrix metabolism and the suppression of VEGFR2 autoactivation. In humans, the mutations in TIMP3 are responsible for Sorsby fundus dystrophy (SFD), an autosomal, dominant, and inherited retinal dystrophy [56,57,58]. Recently, choroid thinning and increased flow signal deficiency in the choriocapillaris have been reported in patients with SFD [58]. It is considered that the degeneration of the choroid in SFD results from the accumulation of mutant TIMP3 protein in the Bruch’s membrane, while it might be worth considering that the developmental anomaly in the choroid is involved in the pathology of SFD.

The importance of RPE for the developing choroidal circulation system is well established, as overviewed above. However, although some studies indicate the window for the RPE dependency of choroidal development [41,42], the roles of RPE in the different timings of development are still unclear. A study focusing on this point would shed light on a novel aspect of the interaction between the RPE and the choroid.

**Table 2 ijms-24-05371-t002:** The summary of model mice for the analyses of the development of choroidal circulation system.

Model Mouse	Genetic Modification	Phenotype	Relevant Disease	Ref
*Tyrp1-Fgf9*	Overexpression in RPE	Lack of RPE Lack of choroid		[40]
*βB1-Tgfb1*	Overexpression in lens	Lack of RPE Choriocapillaris atrophy		[41]
*Best1-rtTA;TRE-Cre;Vegfa* ^flox/flox^	Knockout in RPE (drug-inducible)	Hypoplasia of choriocapillaris Hypoplasia of large vessels		[42]
*Tyrp1-Cre*^tg/0^;*Vegfa*^flox/flox^	Knockout in RPE	Hypoplasia of choriocapillaris		[43]
*Best1-Cre;Vegfa* ^flox/flox^	Knockout in RPE	Hypoplasia of choriocapillaris Elongation of axial length Decreased refraction	Myopia	[44]
*Best1-Cre;Vhl* ^flox/flox^	Knockout in RPE	Vasodilatation		[44]
*Tyrp1-trFgfr1*	Knockout in RPE	Immature choriocapillaris		[39]
*Lrp2* ^267/−^	Heterozygote knockout with the missense mutant at amino acid position 2721 of the LRP2 (line 267)	Enlarged and exophthalmic eyes Thinning of retina Hyperplasia of non-pigmented epithelium of the ciliary body Elongation of axial length	Donnai–Barrow syndrome	[53]
*Best1-Cre;Lrp2* ^flox/flox^	Knockout in RPE	Enlarged eyes Thinning of retina Elongation of axial length Hypoplasia of choriocapillaris Thinning of sclera	Donnai–Barrow syndromeSevere myopia	[44]
*Chx10-Cre;Lrp2* ^flox/flox^	Knockout in neural retina	No phenotype		[44]
*Timp3* KO	Total knockout	Dilated choroidal vessels	Sorsby fundus dystrophy	[55]
*Aldh1a* KO	Total knockout	Hypoplasia of choriocapillaris (Dorsal specific)		[59]
*Dct-Cre;Sox9* ^flox/flox^	Knockout in RPE	Hypoplasia of choriocapillaris		[60]
*Wnt1-Cre;Angpt1* ^flox/flox^	Knockout in neural crest cells	*At birth* Hypoplasia of choriocapillaris, dilated venules Reduction in vortex vein number *1 year of age* Pachyvessels in the choroid RPE dysplasia Subretinal choroidal neovascularization	Polypoidal choroidal vasculopathy Central serous chorioretinopathy	[61]
*Mitf*^mi-bw^/*Mitf*^mi-bw^	Knockout in melanocytes	Normal choriocapillaris Reduced branching of posterior ciliary arteries and collected venules	Waardenburg syndrome type 2	[62]
*Ihh* KO	Total knockout	Misshapen eyes Hypopigmentation in RPE Immature sclera Choroid thinning		[63]

## 8. Retinal Progenitor Cells

In choroidal development, region-specific regulation by retinal progenitor cells is also known [59]. The progenitor cells in the developing dorsal neural retina express *Aldh1a1*, which synthesizes retinoic acids (RA). The *Aldh1a1* KO mice show hypoplasia in the choriocapillaris of the dorsal side of the eyeballs (Table 2). *Vegfa* is downregulated as well as *Sox9*, a positive regulator of *Vegfa* transcription, in the dorsal RPE of *Aldh1a1* KO mice. The prenatal administration of RA to the pregnant *Aldh1a1* KO mice prevent dorsal choroidal hypoplasia in embryos, and RA treatment to *Aldh1a1* KO RPE cells in vitro increases the expression of *Sox9* and *Vegfa*, suggesting that the retinal progenitor cells regulate dorsal choroid development by RA production via stimulation of the *Sox9-Vegfa* axis in the RPE. On the other hand, according to another study, the necessity of *Sox9* expression in the RPE is not limited to dorsal choroidal development. RPE-specific *Sox9* deleted mice (*Dct-Cre;Sox9^flox/flox^* mice) show hypoplasia of the choriocapillaris in regions near the optic nerve and posterior ciliary arteries [60]. In *Dct-Cre;Sox9^flox/flox^* mice, the decreased expression of *Vegfa* is not detected, but another proangiogenic factor *Angptl4* shows decreased expression. Clarifying the common and different roles between the *Sox9-Vegfa* axis and the *Sox9-Angptl4* axis would provide further insight into choroidal development.

## 9. Neural Crest-Derived Stromal Cells

Angiopoietin (ANGPT)-TEK signaling is a well-established regulator of vascular formation [64,65]. Tyrosine kinase endothelial (TEK), also known as Tie2, is a receptor of ANGPT1 and ANGPT2. The binding of ANGPT1 to TEK activates the kinase activity of TEK to stabilize vasculature, but the binding of ANGPT2 fails to activate TEK kinase activity, causing instability and remodeling of the vasculature. In the choroid, *Angpt1* is expressed in neural crest-derived stromal cells [61]. The neural crest-specific *Angpt1* deletion in *Wnt1-Cre;Angpt1^flox/flox^* mice induces hypoplasia of the choriocapillaris, abnormally dilated venules, and a reduction in the vortex vein number at birth. At 1 year of age, they show pachychoroid disease-like symptoms such as pachyvessels in the choroid, RPE dysplasia, and subretinal choroidal neovascularization (Table 2). Thus, ANGPT-TEK signaling has an essential role in developing the choroidal circulation system. In humans, polypoidal choroidal vasculopathy-associated variants are found in *ANGPT2* [66] and *TEK* [67], and additional variants related to central serous chorioretinopathy are identified in *PTPRB* [68], which encodes phosphatase dephosphorylating TEK [69]. Further research about the roles of stromal cells and ANGPT-TEK signaling in choroidal development would contribute to a further understanding of ocular diseases.

## 10. Melanocytes

The role of melanocytes in choroidal development has been reported [62]. *Mitf^mi-bw^/Mitf^mi-bw^* mice lack melanocytes including choroidal melanocytes because they lack the expression of the *Mitf*-M isoform indispensable for melanocyte differentiation, while their RPE is pigmented and has normal functions. The choroid in *Mitf^mi-bw^/Mitf^mi-bw^* mice is thinner and poorly vascularized compared with wild-type mice. Differently from the other models mentioned above, in *Mitf^mi-bw^/Mitf^mi-bw^* mice, the choriocapillaris shows normal morphology, and thinning and poor vascularization is due to reduced branching of posterior ciliary arteries and collected venules (Table 2). Therefore, although the molecular mechanism mediating melanocyte–vascular interaction is not identified, choroidal melanocytes would be required to form the Sattler’s and Haller’s layers. The phenotypes of *Mitf^mi-bw^/Mitf^mi-bw^* mice resemble the symptoms of Waardenburg syndrome (WS) type 2 [70,71]. A thinning choroid has also been reported in some patients with WS [72,73], but the case number is limited. Thus, it is difficult to discuss the relevancy of choroidal hypoplasia in *Mitf^mi-bw^/Mitf^mi-bw^* mice in human ocular disorder. Nonetheless, the phenotype could offer valuable insight into understanding choroidal development.

## 11. The Choroid as a Regulator of Ocular Development

While the development of the choroid is regulated by various types of cells or tissues, as previously described, a study has demonstrated that the RPE and sclera development are regulated by *Ihh* signaling from the choriocapillaris [63]. In mouse embryos after E 11.5, the *Ihh*-expressing cells underlay the RPE. The co-expression of collagen IV (ColIV) and platelet endothelial cell adhesion molecule (PECAM1) in *Ihh-*expressing cells is confirmed at E 14.5, demonstrating that *Ihh*-expressing cells are the endothelial cells of the choriocapillaris. The expression of the *Gli1*-encoding patched (PTC1; a hedgehog receptor) is detected in the RPE, the choroidal endothelial cells, and the mesenchymal cells localized in the outer region of the choroidal endothelial cells. The eyes of *Ihh* KO mice lacking the *Ihh* gene show a hypopigmented RPE and misshapen phenotype at E 18.5 (Table 2). The abnormal eye shape in *Ihh* KO mice is due to the loss of condensed fibers and failure of mature fibroblasts in the sclera. In the ultrastructure, RPE cells fail to touch the neural retinal cells in *Ihh* KO mice. Choroids also become thinner in comparison with wild-type mice. Thus, the development of the RPE, sclera, and the choroid itself requires Indian hedgehog (IHH) secreted from choroidal endothelial cells. This study clearly shows that developing choroid functions regulate the development of other ocular tissues. Although human ocular disorder caused by *Ihh* mutation is not known when we describe this review, disruption in the choroidal regulation of optic tissue development might be involved in human ocular disorders.

## 12. Summary

In this review, we provide an overview of the development of the choroidal circulation system (Table 2 and Figure 3). The development of the choroidal circulation system is regulated by the interaction with the retinal tissues (RPE and retinal progenitor cells) and the choroidal stromal tissues (neural crest-derived stromal cells and melanocytes) mediated by the small molecule and secreting factors such as RA, VEGFA, FGF2, TIMP3, and ANGPT1 (Figure 3). Additionally, the involvement of dysregulations of these interactions in congenital ocular disorders is becoming understood. On the other hand, recent studies reported that 11 and 13 kinds of cells, including the endothelial cell, compose the choroid in mice and humans, respectively [74,75]. We have little information about their roles in developing the choroidal circulation system. Although many other signaling and matrix metabolizing factors regulate vascular development both positively and negatively [54,76], their roles in the developing choroid are also unknown. Light stimulation might also be involved in the development of the choroid, as well as the vitreous and retina [32,33]. Thus, research is just beginning to understand the developmental regulation of the choroidal circulation system. Further studies will not only expand our knowledge of ocular development but will also support the understanding of ocular disorders.

## Figures and Tables

**Figure 1 ijms-24-05371-f001:**
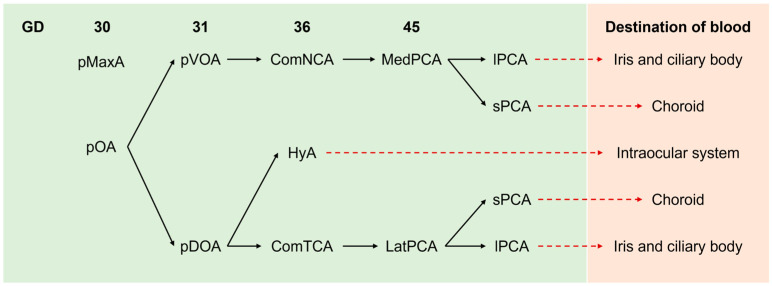
A time series summary of branching of pOA in humans. At 30 GD, pMaxA is responsible for blood supply to the embryonic ocular regions. After 31 GD, pOA and its branches supply blood to developing ocular systems, as indicated. pMaxA: primitive maxillary artery, pOA: primitive ophthalmic artery, ComNCA: common nasal ciliary artery, ComTCA: common temporal ciliary artery, HyA: hyaloidal artery, MedPCA: medial posterior ciliary artery, LatPCA: lateral posterior ciliary artery, lPCA: long posterior ciliary artery, sPCA: short posterior ciliary artery.

**Figure 2 ijms-24-05371-f002:**
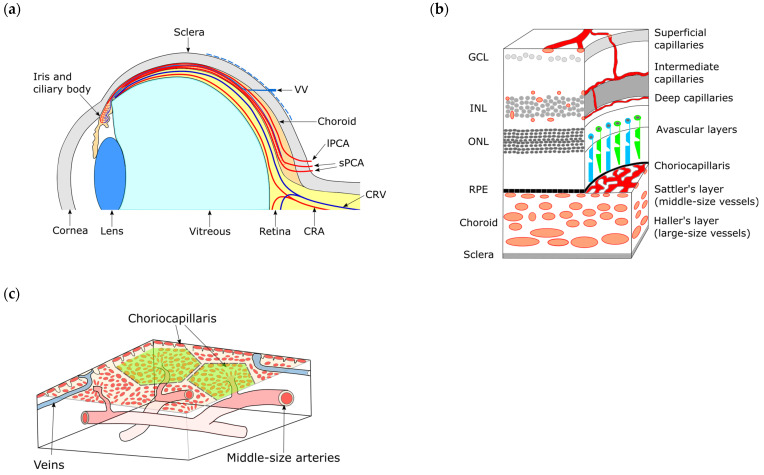
Summary of blood supply in ocular system. (**a**) The blood supply from posterior side of adult ocular system. CRA developed from hyaloid artery is responsible for the intraocular blood supply. sPCAs supply blood to choroid and lPCAs supply blood to iris and ciliary body. The blood supplied by sPCA and lPCA is drained through VV. CRA: central retinal artery, CRV: central retinal vein, sPCA: sort posterior ciliary arteries, lPCA: long posterior ciliary arteries, VV: vortex vein. (**b**) The retinal and choroidal circulation in mature eyes. Blood is supplied by 3 layers of capillary plexus in inner layers of retina. In developing retina, the superficial capillaries are inhibited by the entry into neuroretina to form intermediate and deep capillaries. Choriocapillaris is responsible for the supply of nutrients and gas to avascular layers involving RPE and photoreceptors. GCL: ganglion cell layer, INL: inner nuclear layer, ONL: outer nuclear layer. (**c**) A schema of lobular patterning of choriocapillaris. The lobular patterns are indicated by green polygons.

**Figure 3 ijms-24-05371-f003:**
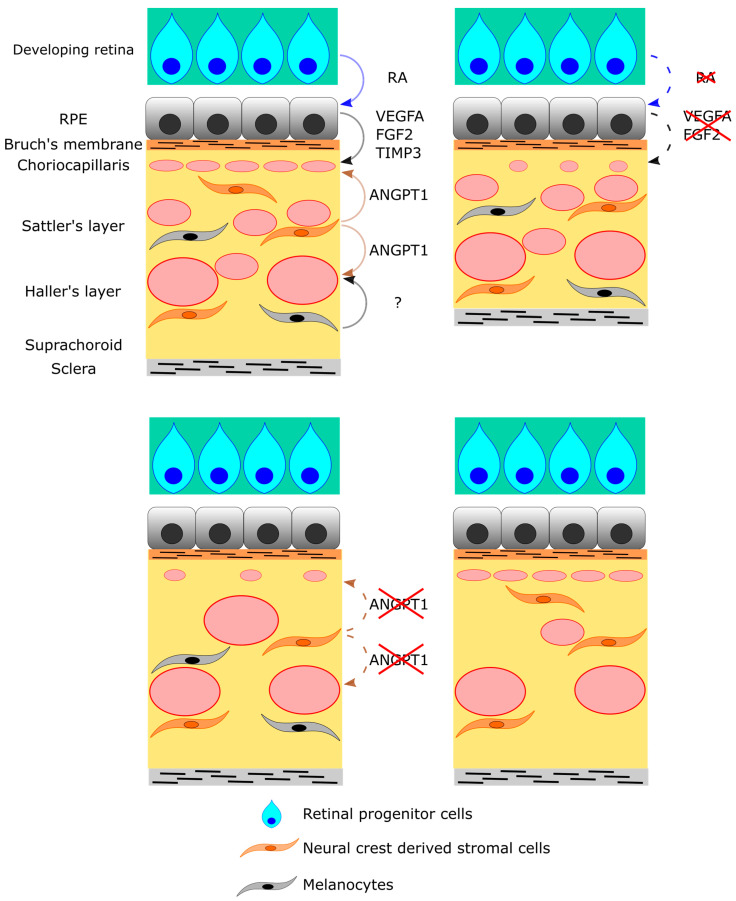
A summary of cellular and molecular interaction in choroid development. As indicated in the upper-left panel, RPE regulates the development of choriocapillaris by the secretion of VEGFA, FGF2, and TIMP3. On the dorsal side of eye, retinal progenitor cells stimulate RPE to produce VEGFA via secretion of RA. Neural crest-derived stromal cells secrete ANGPT1 to regulate the development of choriocapillaris and middle and large vessels. Melanocytes also regulate the development of middle and large vessels, but the molecular mechanism is unknown. The lack of VEGFA or FGF2 from RPE causes hypoplasia of choriocapillaris and choroid thinning (upper-right panel). The deletion of ANGPT1 in neural crest-derived stromal cells results in hypoplasia of choriocapillaris, dilated venules, and reduction in vortex vein number (lower-left panel). The lack of melanocytes induces reduced branches of sPCA and venules but does not affect the development of choriocapillaris (lower-right panel).

**Table 1 ijms-24-05371-t001:** The summary of general development of ocular systems.

Developmental Stages		Events in Each Tissue				
Human	Mouse	Retina	Lens	Cornea	Iris and Ciliary Body	Periocular Organs
~3 WG	~E 8.25	Optic sulcus				
~3.5 WG	~E 8.5	Optic vesicle	Lens placode			
~4 GW	~E 9.5					
~5 GW	~E 10	Optic cup	Lens pit			
~6 GW	~E 11.5					Eyelid formation
~6.5 GW	~E 13.5					Lacrimal grand formation
~10 GW	~E 14.5	Neural retina and RPE	Lens vesicle	Endothelium formation		
~15 GW						Lacrimal grand maturation (human)
~16 GW	~E 15.5			Epithelium formation		
~18 GW	~E 17				Formation of both tissues	
~36 GW						Eyelid separation (human)
	Birth					Lacrimal grand maturation (mouse)
	~Day 12					Eyelid separation (mouse)

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
