# Peer review of "Molecular and Cellular Regulations in the Development of the Choroidal Circulation System"

_ijms, 2023, doi:10.3390/ijms24065371_

Round 1
Reviewer 1 Report
The choriocapillaris endothelial cells are special which have fenestration only on their side facing the RPE. RPE cells play important role in the maintenance of the fenestration choriocapillaris, which is strong modulated by VEGF secreted by RPE cells. VEGFR-2 and VEGFR-3 preferentially localized at the side of the choriocapillaris endothelium facing the RPE cell layer. The authors should add these informations into their paper to make the contents more comprehensive.
Author Response
We appreciate the time and effort you have dedicated to providing insightful feedback on ways to strengthen our paper. We have incorporated changes that reflect the detailed suggestions you have provided. We also hope that our edits and the responses we provide in the attachment satisfactorily address all the issues and concerns you have noted. Please see the attachment for the point-by-point responses.

Reviewer 2 Report
This manuscript, by Imanishi et al., written as a review, discusses the in-utero development of the choroidal circulatory system, from an anatomical, molecular, and cellular signaling perspective.
The subject matter is somewhat novel, as the authors point out. The manuscript is quite well written, but would benefit greatly from more extensive use of pictures and diagrams. Also, the standard of written English, although quite good, is not of publication quality. Unfortunately, there are many grammatical and tense errors throughout. If publishing in an English-language journal, the entire manuscript needs to be reviewed by a native English speaker or a professional medical writing service.
2. General Development of the Ocular System
A diagram (or several diagrams) showing the development of the eye as discussed in the text would be of great value, especially for those less familiar with the subject matter.
(Lines 55-56). The mesenchymal cells touching the lens vesicle change to epithelium-like morphology and form corneal endothelium. Be careful with definitions here. Some would argue that epithelial tissue is derived from embryonic ectoderm or embryonic endoderm, whereas endothelial tissue is derived from embryonic mesoderm. Therefore, the statement that mesenchymal cells change to an epithelium-like morphology (as opposed to an endothelium-like morphology) is debatable. The authors should include various viewpoints on this issue and include supporting references.
(Lines 60-61). Are the authors able to cite a reference which states that the iris stroma and ciliary muscle is derived from mesenchyme? Should this not say neural crest, which is in turn derived from mesenchyme? Please clarify.
3. The Embryonic Blood Supply to the Eyes
Once again, a diagram would be very helpful to the reader here.
(Line 115). The Long Posterior Ciliary Artery supplies blood to the ciliary body and iris, mainly, in the mature eye. Is this different during the embryologic stage?
5. Anatomy of the Choroid
(Line 142). Bruch’s membrane is considered to consist of 5 layers:
The Basement membrane of the Retinal Pigment Epithelium
An Inner Collagenous Zone
A Central Band of Elastic Fibers
An Outer Collagenous Zone
The Basement Membrane of the Choriocapillaris
Thus, Bruch’s membrane is arguably a part of the retina as well as a part of the choroid. Or perhaps also something in between. To simply state that it is a part of the choroid is a little misleading.
(Line 153). When the authors say, ‘like erythroblasts’, do they mean to imply that these cells have some erythroblast-like qualities, or do they mean these cells are erythroblasts, or do they mean these cells are a mixture of erythroblasts and some other cell types. If the latter, please specify the other cell types involved.
7. RPE Cells. Please reference Table 1 in the text somewhere here.
8. Retinal Progenitor Cells. Please reference Table 1 in the text somewhere here.
9. Neural Crest Derived Stromal Cells. Please reference Table 1 in the text somewhere here.
10. Melanocytes (Lines 269-271). I do not understand the meaning of the second sentence under this header. Please clarify.
11. Choroid as Regulator of Ocular Development
Reference Table 1 in the text somewhere here.
References. Is citation #43 a book or a monograph? Does it have an ISBN number?
Author Response

(The authors gave the same response as above.)

Round 2
Reviewer 2 Report
The authors have made an excellent job of addressing my questions and concerns. The manuscript is now greatly improved. Thank you very much for taking the time to do this.
Author Response
It is great pleasure that our responses addressed your questions and concerns. We appreciate the time and effort you have dedicated to providing insightful feedback on ways to strengthen our paper.